# Training instance segmentation neural network with synthetic datasets for crop seed phenotyping

Yosuke Toda [1,2✉], Fumio Okura[1,3], Jun Ito[4], Satoshi Okada[5], Toshinori Kinoshita [2], Hiroyuki Tsuji[4] & Daisuke Saisho[5]

In order to train the neural network for plant phenotyping, a sufficient amount of training data must be prepared, which requires time-consuming manual data annotation process that often becomes the limiting step. Here, we show that an instance segmentation neural network aimed to phenotype the barley seed morphology of various cultivars, can be sufficiently trained purely by a synthetically generated dataset. Our attempt is based on the concept of domain randomization, where a large amount of image is generated by randomly orienting the seed object to a virtual canvas. The trained model showed 96% recall and 95% average Precision against the real-world test dataset. We show that our approach is effective also for various crops including rice, lettuce, oat, and wheat. Constructing and utilizing such synthetic data can be a powerful method to alleviate human labor costs for deploying deep learning-based analysis in the agricultural domain.

[1] Japan Science and Technology Agency, 4-1-8 Honcho, Kawaguchi, Saitama 332-0012, Japan. [2] Institute of Transformative Bio-Molecules (WPI-ITbM), Nagoya University, Chikusa, Nagoya 464-8602, Japan. [3] Department of Intelligent Media, Institute of Scientific and Industrial Research, Osaka University, 8-1 Mihogaoka, Ibaraki, Osaka 567-0047, Japan. [4] Kihara Institute for Biological Research, Yokohama City University, Maioka 641-12, Totsuka, Yokohama 244-0813, Japan. [5] Institute of Plant Science and Resources, Okayama University, Chuo 2-20-1, Kurashiki, Okayama 710-0046, Japan.
✉email: tyosuke@aquaseerser.com

Deep learning[1] has gathered wide attraction in both the scientific and industrial communities. In computer vision field, deep-learning-based techniques using convolutional neural network (CNN) are actively applied to various tasks, such as image classification[2], object detection[3,4], and semantic/instance segmentation[5–7]. Such techniques have also been influencing the field of agriculture. This involves image-based phenotyping, including weed detection[8], crop disease diagnosis[9,10], fruit detection[11], and many other applications as listed in the recent review[12]. Meanwhile, not only features from images but also with that of environmental variables, functionalized a neural network to predict plant water stress for automated control of greenhouse tomato irrigation[13]. Utilizing the numerous and high-context data generated in the relevant field seems to have high affinity with deep learning.

However, one of the drawbacks of using deep learning is the need to prepare a large amount of labeled data. The ImageNet dataset as of 2012 consists of 1.2 million and 150,000 manually classified images in the training dataset and validation/test dataset, respectively[14]. Meanwhile, the COCO 2014 Object Detection Task constitutes of 328,000 images containing 2.5 million labeled object instances of 91 categories[15]. This order of annotated dataset is generally difficult to prepare for an individual or a research group. In the agricultural domain, it has been reported that sorghum head detection network can be trained with a dataset consisting of 52 images with an average of 400 objects per image[16], while a crop stem detection network was trained starting from 822 images[17]. These case studies imply that the amount of data required in a specialized task may be less compared with a relatively generalized task, such as ImageNet classification and COCO detection challenges. Nonetheless, the necessary and sufficient amount of annotation data to train a neural network is generally unknown. Although many techniques to decrease the labor cost, such as domain adaptation or active learning, are widely used in plant/bio science applications[18–20], the annotation process is highly stressful for researchers, as it is like running a marathon without knowing the goal.

A traditional way to minimize the number of manual annotations is to learn from synthetic images, which is occasionally referred to as the sim2real transfer. One of the important advantages of using a synthetic dataset for training is that the ground-truth annotations can be automatically obtained without the need for human labor. A successful example can be found in person image analysis method that uses the image dataset with synthetic human models[21] for various uses such as person pose estimation[22]. Similar approaches have also been used for the preparation of training data for plant image analysis. Isokane et al.[23] used the synthetic plant models for the estimation of branching pattern, while Ward et al. generated artificial images of Arabidopsis rendered from 3D models and utilized them for neural network training in leaf segmentation[24].

One drawback of the sim2real approach are the gaps between the synthesized images and the real scenes, e.g., nonrealistic appearances. To counter this problem, many studies attempt to generate realistic images from synthetic datasets, such as by using generative adversarial networks (GAN)[25,26]. In the plant image analysis field, Giuffrida et al.[27] used GAN-generated images to train a neural network for Arabidopsis leaf counting. Similarly, Arsenovic et al. used StyleGAN[28] to create training images for the plant disease image classification[29].

On the other hand, an advantage of sim2real approach is the capability of creating (nearly) infinite number of training data. Approaches that are bridging the sim2real gap by leveraging the advantage is domain randomization, which trains the deep networks using large variations of synthetic images with randomly sampled physical parameters. Although domain randomization is somewhat related to data augmentation (e.g., randomly flipping and rotating the images), the synthetic environment enables the representation of variations under many conditions, which is generally difficult to attain by straightforward data augmentation techniques for real images. An early attempt at domain randomization was made by generating the images using different camera positions, object location, and lighting conditions, which is similar to the technique applied to control robots[30]. For object recognition tasks, Tremblay et al.[31] proposed a method to generate images with a randomized texture on synthetic data. In the plant-phenotyping field, recently, Kuznichov et al. proposed a method to segment and count leaves of not only Arabidopsis, but also that of avocado and banana, by using a synthetic leaf texture located with various size/angles, so as to mimic images that were acquired in real agricultural scenes[32]. Collectively, the use of synthetic images has a huge potential in the plant-phenotyping research field.

Seed shape, along with seed size, is an important agricultural phenotype. It consists of yield components of crops, which are affected by environmental condition in the later developmental stage. The seed size and shape can be predictive on germination rates and subsequent development of plants[33,34]. Genetic alteration of seed size contributed a significant increase in thousand-grain weight in contemporary barley-cultivated germplasm[35]. Several studies report the enhancement of rice yield by utilizing seed width as a metric[36,37]. Moreover, others utilized elliptic Fourier descriptors that enable to handle the seed shape as variables representing a closed contour, successfully characterizing the characters of various species[38–41]. Focusing on morphological parameters of seeds seems to be a powerful metric for both crop-yield improvement and for biological studies. However, including the said reports, many of the previous studies have evaluated the seed shape by qualitative metrics (e.g., whether the seeds are similar to the parental phenotype), by vernier caliper, or by manual annotation using an image- processing software. The phenotyping is generally labor-intensive and cannot completely exclude the possibility of quantification errors that differ by the annotator. To execute a precise and large-scale analysis, automation of the seed-phenotyping step was preferred.

In recent years, several studies have been reported to systematically analyze the morphology of plant seeds by image analysis. Ayoub et al. focused on barley seed characterization in terms of area, perimeter, length, width, F circle, and F shape based on digital camera-captured images[42]. Herridge et al. utilized a particle analysis function of ImageJ (https://imagej.nih.gov/ij/) to quantify and differentiate the seed size of Arabidopsis mutants from the background population[43]. SmartGrain software has been developed to realize the high-throughput phenotyping of crop seeds, successfully identifying the QTL that is responsible for seed length of rice[44]. Miller et al. reported a high-throughput image analysis to measure morphological traits of maize ears, cobs, and kernels[45]. Wen et al. developed an image analysis software that can measure seed shape parameters such as width, length, and projected area, as well as the color features of maize seeds: they found a correlation between these physical characteristics with seed vigor[46]. Moreover, commercially available products such as Germination Scanalyzer (Lemnatec, Germany) and PT portable tablet tester (Greenpheno, China) also aim or have the ability to quantify the morphological shape of seeds. However, the aforementioned approaches require the seeds to be sparsely oriented for efficient segmentation. When seeds are densely sampled and physically touching each other, they are often detected as a unified region, leading to an abnormal seed shape output. This requires the user to manually reorient the seeds in a sparse manner, which is a potential bar to secure sufficient amount of biological replicate in the course of high-throughput analysis. In

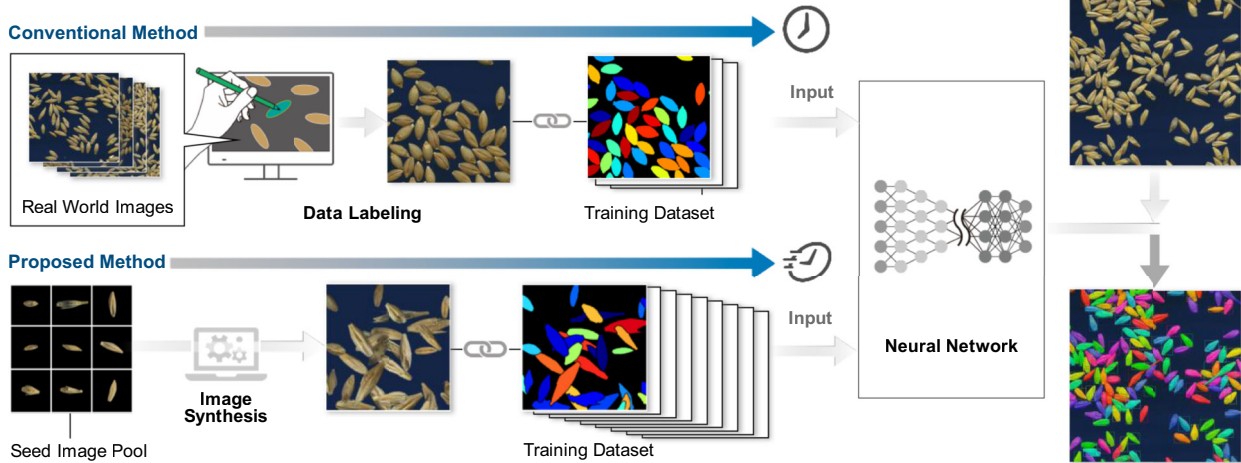

**Fig. 1 Overview of the proposed training process of crop seed instance segmentation.** Conventional method requires manual labeling of images to generate the training dataset, while our proposed method can substitute such step by using a synthetic dataset for crop seed instance segmentation model.

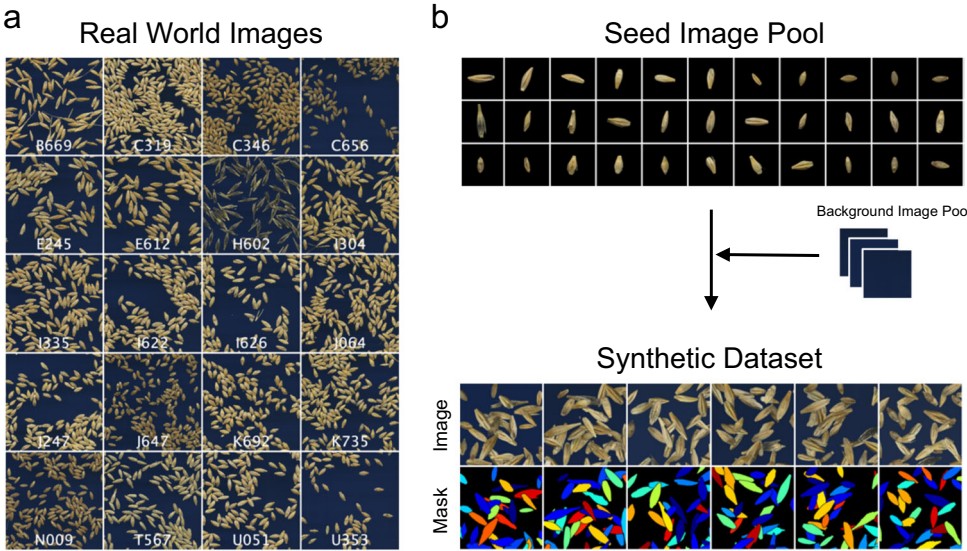

**Fig. 2 Data prepared in this study. a** Images of barley seeds scanned from 20 cultivars. Cultivar names are described in white text in each image. These images were also used as a real-world test dataset in Table 1. **b** Scheme of generating synthetic images. Images are generated by combining actual scanned seed images over the background images onto the virtual canvas. Simultaneously generated ground-truth label (mask) is shown at the bottom, in which each seed area is marked with a unique color.

such situations, deep-learning-based instance segmentation can be used to overcome such a problem by segmenting the respective seed regions regardless of their orientation. Nonetheless, the annotation process as described previously was thought to be the potential limiting step.

In this paper, we show that utilizing a synthetic dataset that the combination and orientation of seeds are artificially rendered, is sufficient to train an instance segmentation of a deep neural network to process real-world images. Moreover, applying our pipeline enables us to extract morphological parameters at a large scale with precise characterization of barley natural variation at a multivariate perspective. The proposed method can alleviate the labor-intensive annotation process to realize the rapid development of deep-learning-based image analysis pipeline in the agricultural domain as illustrated in Fig. 1. Our method is largely related to the sim2real approaches with the domain randomization, where we generate a number of training images by randomly locating the synthetic seeds with actual textures by changing its orientation and location.

The contribution of this study is twofold. First, this is the first attempt to utilize a synthetic dataset (i.e., a sim2real approach) with domain randomization for the crop seed phenotyping, which can significantly decrease the manual labor for data creation (Fig. 1). Second, we propose a first method that can be used against the densely sampled (e.g., physically touching) seeds using instance segmentation.

## Results

**Preparation of barley seed synthetic dataset**. Examples of seed images captured by the scanner are shown in Fig. 2a. The morphology of barley seeds is highly variable between cultivars, in terms of size, shape, color, and texture. Moreover, the seeds randomly come in contact with or partially overlap each other. Determination of the optimal threshold for binarization may enable isolation of the seed region from the background; however, conventional segmentation methods such as watershed require extensive search for suitable parameters per cultivar to efficiently

segment the single-seed area for morphological quantification. Establishment of such pipeline requires an extensive effort of an expert. Employing a sophisticated segmentation method (in our case, instance segmentation using Mask R-CNN[7]) is indeed a choice for successful separation of the individual seeds. However, Mask R-CNN requires annotations of bounding boxes—which circumscribe the seed—and mask images that necessarily and sufficiently cover the seed area (Supplementary Fig. 1). Given that the numbers of seeds per image are abundant (Fig. 2a), the annotation process has been predicted to be labor-intensive.

Figure 2b shows the seed image pool and synthesized dataset obtained using the proposed method (see "Methods" for details). Instead of labeling real-world images for use as a training dataset, Mask R-CNN was trained using the synthetic dataset (examples shown at the bottom of Fig. 2b), which is generated from the seed and background image pool (Fig. 2b top) using a domain randomization technique.

**Model evaluation**. We show herein the visual results and a quantitative evaluation of object detection and instance segmentation by Mask R-CNN. The trained Mask R-CNN model outputs a set of bounding box coordinates and masks images of seed regions (raw output) (Fig. 3a, top row). Examples of visualized raw output obtained from the real-world images show that the network can accurately locate and segment the seeds regardless of their orientation (Fig. 3b; Supplementary Fig. 2). Table 1 summarizes the quantitative evaluation using the recall and AP measures (see "Methods" for details). The efficacy of seed detection was evaluated using the recall values computed for bounding box coordinates at 50% Intersection of Union (IoU) threshold ($Recall_{50}$). The model achieved an average of 95 and 96% on the synthetic and real-world test datasets, respectively. This indicates that the trained model can locate the position of seeds with very low false-negative rate. From the average precision (AP) values, which were computed based on mask regions at varying mask IoU thresholds, comparable $AP_{50}$ values were achieved between the synthetic (96%) and real-world (95%) datasets. For higher IoU threshold (AP@ [.5:.95] and $AP_{75}$), the values of the synthetic test dataset (73%) exceeded that of the real-world test dataset (59%). These results suggest that the model's ability to segment the seed region is better in the case of the synthetic than the real-world images. The higher values in the synthetic dataset possibly derive from data leak, which the same seed images appear as in the training dataset, but even the orientation and combination of seeds area are different. However, considering the visual output interpretation (Fig. 3b) and the values of $AP_{50}$ (95%) in the real-world test dataset, we judged that seed morphology can be sufficiently determined from real-world images. The relatively low AP in high IoU in the real-world test dataset is possibly derived from the subtle variation in the manual annotation of seed mask regions. It is noteworthy that when the Mask R-CNN model was trained with the manually annotated seeds, the network showed poor performance in segmenting the seed regions (Supplementary Fig. 3). This was especially apparent when the seeds were physically touching each other and forming a dense cluster, which further supports the efficiency of domain randomization.

**Post processing**. As described in the Methods section, we introduced a post-processing step to the raw output to eliminate detections that are not suitable for further analysis. This process removes seed occlusion due to physical overlap, incomplete segmentation by the neural network, non-seed objects such as dirt or awn debris, or the seeds that were partly hidden due to the location being outside the scanned area (Fig. 3c). Figure 3d shows

the distribution of the seed area before and after post processing. Even though the seed area itself was not used as a filtering criterion, the area values in the respective cultivars shift from a long-tailed to a normal distribution, which well reflects the characteristics of a homogenous population (Fig. 3d). A comparison of the filtered output (inferenced seed area) and hand-measured (ground-truth area) values displays a strong correlation, where the Pearson correlation value is 0.97 (Fig. 3e). These results suggest that the filtered output values obtained from our pipeline are reliable for further phenotypic analyses.

**Morphological characterization of barley natural variation**. Our pipeline learns from synthetic images, which ease the training dataset preparation process. This pipeline enables large-scale analysis across multiple cultivars or species. To highlight the important advantages of the proposed pipeline, we herein demonstrate an array of analyses to morphologically characterize the natural variation of barley seeds, which highlights the crucial biological features that will provide guidance for further investigation. We selected 19 out of 20 cultivars that were used to train the neural network; however, we have acquired a new image that was not used for training or testing in further analysis. One accession, H602, was excluded from the analysis because the rachis could hardly be removed by husk threshing; therefore, the detected area did not reflect the true seed shape. From the pipeline, we obtained 4464 segmented seed images in total (average of 235 seeds per cultivar).

As simple and commonly used morphological features, the seed area, width, length, and length-to-width ratio per cultivar were extracted from the respective images and are summarized in Fig. 4a–d. With a sufficient number of biological replicates, we can not only compare the inter-cultivar difference (e.g., median or average) but also consider the intra-cultivar variance. We applied the analysis of variance (ANOVA) with Tukey's post hoc test to calculate the statistical difference between cultivars. Many cultivars that visually display similar distribution patterns or medians were grouped into statistically different clusters (e.g., K735 and K692 in Fig. 4a). To gain further insight into the morphology of barley cultivars characterized by various descriptors, we performed a multivariate analysis.

First, we show the results of a principal component analysis (PCA) using eight predefined descriptors (area, width, length, length-to-width ratio, eccentricity, solidity, perimeter length, and circularity). The first two principal components (PC) could explain 88.5% of the total variation (Fig. 5a, b). Although there were no discrete boundaries, the data points tended to form a cluster unique to the cultivar in the latent space, indicating that cultivars can be classified to a certain extent according to the said descriptors (Fig. 5a). Variables such as seed length (L) and perimeter length (PL) mainly constituted the first PC, with seed circularity (CS) oriented toward the opposite direction, while seed width (W) and length-to-width ratio had a major influence on PC2 (Fig. 5b). This is exemplified by the distribution of the slenderest B669 and the circular-shaped J647 at the far-right and far-left orientation in the latent space. Notably, while width (W) mainly constituted PC2, the direction of its eigenvector differs from that of length (L). The moderate value of Pearson's correlation between length and width (0.5, $p < 0.01$) (Supplementary Fig. 4), also implies that genes that control both or either of size and length may coexist in the determination of barley seed shape, as reported in rice[47].

Next, we extracted the contour shapes of seeds using elliptic Fourier descriptors (EFDs) followed by PCA (Fig. 5b, c), which is also used in other studies for seed morphological analysis[38,39]. Compared with the PCA based on the eight morphological

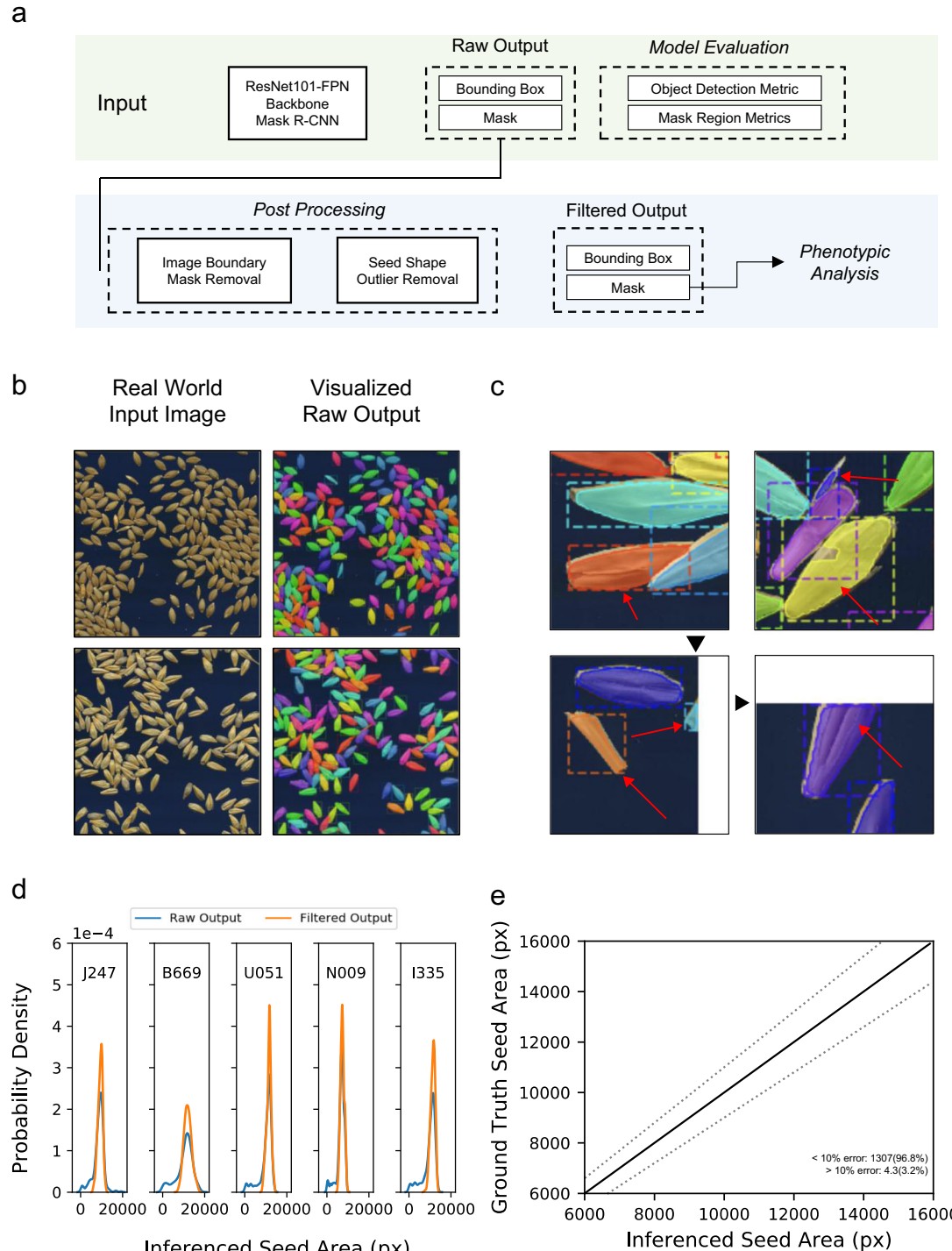

**Fig. 3 Image analysis pipeline. a** Summary of the image analysis pipeline. **b** Examples of the graphical output of the trained Mask R-CNN on real-world images. Different colors indicate an individual segmented seed region. Note that even if the seeds are overlapping or touching each other, the network can discriminate them as an independent object. **c** Examples of detected candidate regions to be filtered in the post-processing step indicated in red arrows. Black arrowheads indicate the input image boundary. **d** Probability density of the seed areas of the raw and filtered output. **e** Scatterplot describing the correlation of the seed area that was measured by the pipeline (inferenced seed area) and by manual annotation (ground-truth seed area). Each dot represents the value by a single seed. Black and gray lines indicate the identity and the 10% error threshold line, respectively. The proportion of the seeds that have lower or higher than the 10% error is also displayed.

descriptors in Fig. 5a, the distributions of the respective seeds were relatively condensed, while the clusters by cultivars were intermixed (Fig. 5c), possibly because the size information is lost upon normalization; therefore, EFD can utilize only the contour shape. Interpolating the latent space in the PC1 axis direction

clearly highlights the difference in slenderness of the seed (Fig. 5d; Fig. Supplementary Fig. 5a). PC2 did not show an obvious change in shape when compared with PC1 (Fig. 5d); however, it seemed to be involved in the sharpness of the edge shape in the longitudinal direction (Supplementary Fig. 5a). Although further

**Table 1 Model evaluation. Table describing the evaluation result of the trained Mask R-CNN raw output.**

| Object detection metric | | Mask region metrics | | |
| --- | --- | --- | --- | --- |
| | Recall$_{50}$ | AP@[.5:.95] | AP$_{50}$ | AP$_{75}$ |
| Synthetic test dataset | 0.95 | 0.73 | 0.96 | 0.93 |
| Real-world test dataset | | | | |
| B669 | 0.92 | 0.56 | 0.92 | 0.84 |
| C319 | 0.95 | 0.62 | 0.91 | 0.86 |
| C346 | 0.98 | 0.64 | 0.97 | 0.89 |
| C656 | 0.96 | 0.61 | 0.95 | 0.92 |
| E245 | 0.95 | 0.63 | 0.94 | 0.84 |
| E612 | 0.96 | 0.66 | 0.98 | 0.89 |
| H602 | 0.87 | 0.42 | 0.78 | 0.41 |
| I304 | 0.99 | 0.64 | 0.98 | 0.88 |
| I335 | 0.97 | 0.67 | 0.93 | 0.92 |
| I622 | 0.93 | 0.62 | 0.93 | 0.87 |
| I626 | 0.96 | 0.65 | 0.95 | 0.89 |
| J064 | 0.93 | 0.65 | 0.97 | 0.86 |
| J247 | 0.94 | 0.65 | 0.97 | 0.86 |
| J647 | 0.98 | 0.62 | 0.98 | 0.92 |
| K692 | 0.98 | 0.69 | 0.98 | 0.93 |
| K735 | 0.95 | 0.62 | 0.92 | 0.86 |
| N009 | 0.99 | 0.63 | 0.99 | 0.91 |
| T567 | 0.98 | 0.63 | 0.98 | 0.88 |
| U051 | 0.96 | 0.65 | 0.96 | 0.89 |
| U353 | 1.00 | 0.65 | 0.98 | 0.89 |
| Average | 0.96 | 0.59 | 0.95 | 0.86 |

Recall values at the IoU threshold of 50% (Recall$_{50}$) and average precision (AP) at the IoU 50% (AP$_{50}$), 75% (AP$_{75}$), and the mean value from IoU 50 to 95% with the step size of 5% (AP@[.5:.95]) are shown.

verification is required, rendering the average contours that represent the shapes of the respective cultivars implies the difference in such metrics (Supplementary Fig. 5b).

Finally, we trained a variational autoencoder (VAE) for latent space visualization[48]. Unlike other methods using the shape descriptors (i.e., eight simple features or EFDs), the VAE inputs the segmented seed images, which can thus obtain a representation that well describes the dataset without feature predefinition (see Methods for details). We have expected that such neural networks can learn the high-level feature (complex phenotype) such as textures, in addition to contour shape and morphological parameters we have handled in Fig. 5a–d. The learned representation can be visualized into a two-dimensional scatterplot similar to a PCA (Fig. 5e). Compared with the PCA-based methods, VAE seems to cluster the cultivar in the latent space more explicitly. While the predefined morphological descriptors extract a limited amount of information from an image, VAE can handle an entire image itself; hence, the latter theoretically can learn more complex biological features. Overall, Z1 tends to be involved in the seed color (i.e., brightness) and size, while Z2 is in seed length (Fig. 5f). These results suggest the potential power of utilizing deep learning for further phenotypic analysis, in addition to the well-established morphological analysis.

**Application in various crop seeds**. We further extended our method to verify the efficacy of our approach for other crop seeds. In this report, we newly trained our model to analyze the seed morphology of wheat, rice, oat, and lettuce, with the, respectively, generated synthetic datasets (Fig. 6, top row). Processing the real-world images resulted in a clear segmentation of each species, regardless of seed size, shape, texture or color, and background (Fig. 6, middle and bottom rows). In conclusion, these results

strongly suggest the high generalization ability of our presented method.

## Discussion

In this research, we showed that utilizing a synthetic dataset can successfully train the instance segmentation neural network to analyze the real-world images of barley seeds. The values obtained from the image analysis pipeline were comparable to that of manual annotation (Fig. 3e), thus achieving high-throughput quantification of seed morphology in various analyses. Moreover, our pipeline requires a limited number of synthesized images to be added to the pool for creating a synthetic dataset. This is labor cost-efficient and practical compared with labeling numerous amounts of images required for deep learning.

To completely understand the use of synthetic data for deep learning, we must have a precise understanding of "what type of features are critical to represent the real-world dataset". In the case of seed instance segmentation, we presumed that the network must learn the representation that is important for segregating physically touching or overlapping seeds into an individual object. Therefore, in the course of designing synthetic images, we prioritized the dataset to contain numerous patterns of seed orientation, rather than to contain massive patterns of seed textures. Based on the result that the model showed sufficient result against the test dataset (Fig. 3b; Supplementary Fig. 2, Table 1), it is suggested that our presumption was legitimate to a certain extent. However, because the neural network itself is a black box, we cannot discuss more than ex post facto reasoning. Recently, there have been challenges to understand the representation of biological context by various interpretation techniques[10,49]. Extending such approaches applicable to an instance segmentation neural network as used in our study will help verify the authenticity of both the synthesized dataset and the trained neural network in future studies.

Notably, it is expected that the model performance will be greatly influenced by the image resolution and variance of seed images used to create the synthetic image, as well as the number of images that constitute the training dataset. Optimal parameters will also depend on the type of cultivars that constitute the test dataset. In this study, we used a fixed condition for synthetic dataset generation, in order to prioritize or demonstrate the effectiveness of domain randomization for seed phenotyping. However, in practical situations where the respective users build and execute a customized pipeline, parameter search may benefit them by providing minimal dataset requirement that leads to calculation cost efficiency. Moreover, introducing additional image augmentation techniques in the synthetic dataset such as random color shift and zoom will lead to a more robust model.

We introduced post processing to exclude nonintegral mask regions prior to phenotypic analysis (Fig. 3a, bottom row and Fig. 4c, d). Theoretically, if we can add a category label to the synthetic dataset to determine whether the respective regions are suitable for analysis, the neural network may acquire the classification ability to discriminate such integrity. However, the complexity of synthetic data generation increases, and misdetected or incomplete mask regions cannot be excluded. We presume that heuristic-based post processing is a simple yet powerful approach. Nonetheless, our outlier removal process is based on the assumption that the seed population is homogeneous. It is important to verify if such filtering is valid against the heterogeneous population. Notably, SmartGrain also introduces a post-processing step, involving a repetitive binary dilation and erosion. Those processes were reported to be effective in analyzing the progenies of two cultivars in rice upon QTL analysis[44]. As the post processing is independent of the neural

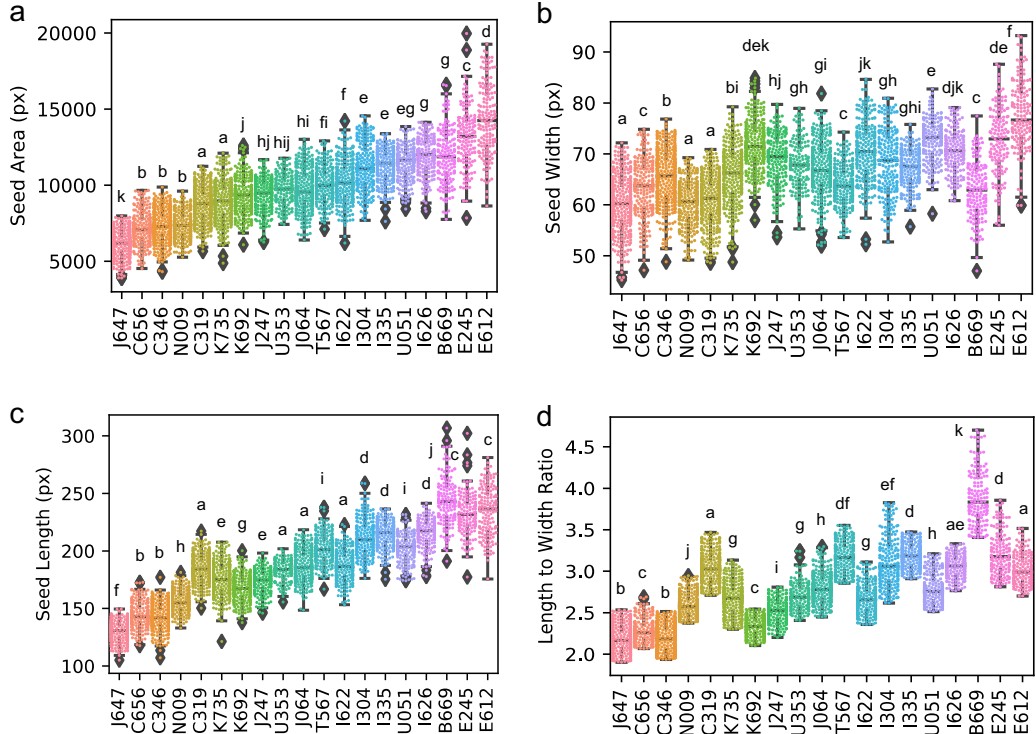

**Fig. 4 Analysis of natural variation of barley seed morphology.** Whisker plot overlaid with a swarm plot (colored dot) grouped by barley cultivars. **a** Seed area, **b** seed width, **c** length, and **d** length-to-width ratio. Diamonds represent outliers. Statistical differences were determined by one-way ANOVA followed by Tukey post hoc analysis. Different letters indicate significant differences ($p < 0.05$).

network in our pipeline, designing and verifying various methods are important for expanding the functionality of the analysis pipeline.

The shape and size of seeds (grains) are important agronomic traits that determine the quality and the yield of crops[50]. In recent years, a number of genes have been identified and characterized through genetic approach, accompanied by laborious phenotyping. In previous studies, researchers manually measured the shape and size of seeds, which is time-consuming and erroneous; it restricted the number of seeds that the researcher can analyze. The researchers used to manually select several seeds that seemed to represent the population in a subjective manner, and for this reason, small phenotypic differences between genotypes could not be detected. Our pipeline can phenotype a large number of seeds without the need to consider the seed orientation to be sparse in image acquisition and thereby can obtain large amount of data in a short period of time. This allows easy and sensitive detection of both obvious and subtle phenotypic differences between cultivars supported by statistical verification (Fig. 4) or by dimensionality reduction methods of multivariate parameters introduced herein (Fig. 5a–d). Moreover, VAE, which requires a sufficient amount of data to fully exert its power to learn the representation of the dataset, becomes also applicable with the data obtained by our approach (Fig. 5e, f). The large-scale analysis across various cultivars provides researchers with yet another option to execute such analyses as demonstrated. This will be a breakthrough in identifying agronomically important genes, especially for molecular genetic research such as genome-wide association study (GWAS), quantitative trait locus (QTL) analysis, and mutant screening. Thus, it will open a new path to identify genes that were difficult to isolate by conventional approaches.

Moreover, the application of our pipeline is not restricted to barley, but can be extended to various crops such as seeds of wheat, rice, oats, and lettuce (Fig. 6). Our results strongly suggest that our approach is applicable to any varieties or species in principle; thus, it is expected to accelerate research in various fields with similar laborious issues. One example can be an application in characterization and gene isolation from seeds of wild species. Cultivated lines possess limited genetic diversity due to bottlenecks in the process of domestication and breeding; therefore many researchers face challenges to identify agronomically important genes from wild relatives as a source of genes for improving agronomic traits. As the appearance of the seeds of wild species is generally more diverse than that of cultivated varieties, development of a universal method to measure both traits was difficult. Another example can be analyzing undetached seeds of small florets (e.g., wheat). Although the shapes of small florets can be manually quantified from the image of a scanned spikelet, the automated quantification has not been realized owing to excess non-seed objects (e.g., glume, awn, and rachis) in the image. Applying another domain of randomization for synthesizing a training dataset can be utilized to functionalize a neural network to quantify seed phenotype from such images.

Collectively, we have shown the efficacy of utilizing the synthetic data, based on the concept of domain randomization to train the neural network for real-world tasks. Recent technical advances in the computer vision domain have enabled us to generate a realistic image, or even a realistic "virtual reality" environment; thus, they will provide more possibilities to give solutions to current image analysis that involved challenges in the agricultural domain. We envision that a collaboration with plant and computer scientists will open a new point of view for generating a workflow that is valuable for plant phenotyping, leading to a further understanding of the biology of plants through the complete use of machine learning/deep-learning methods.

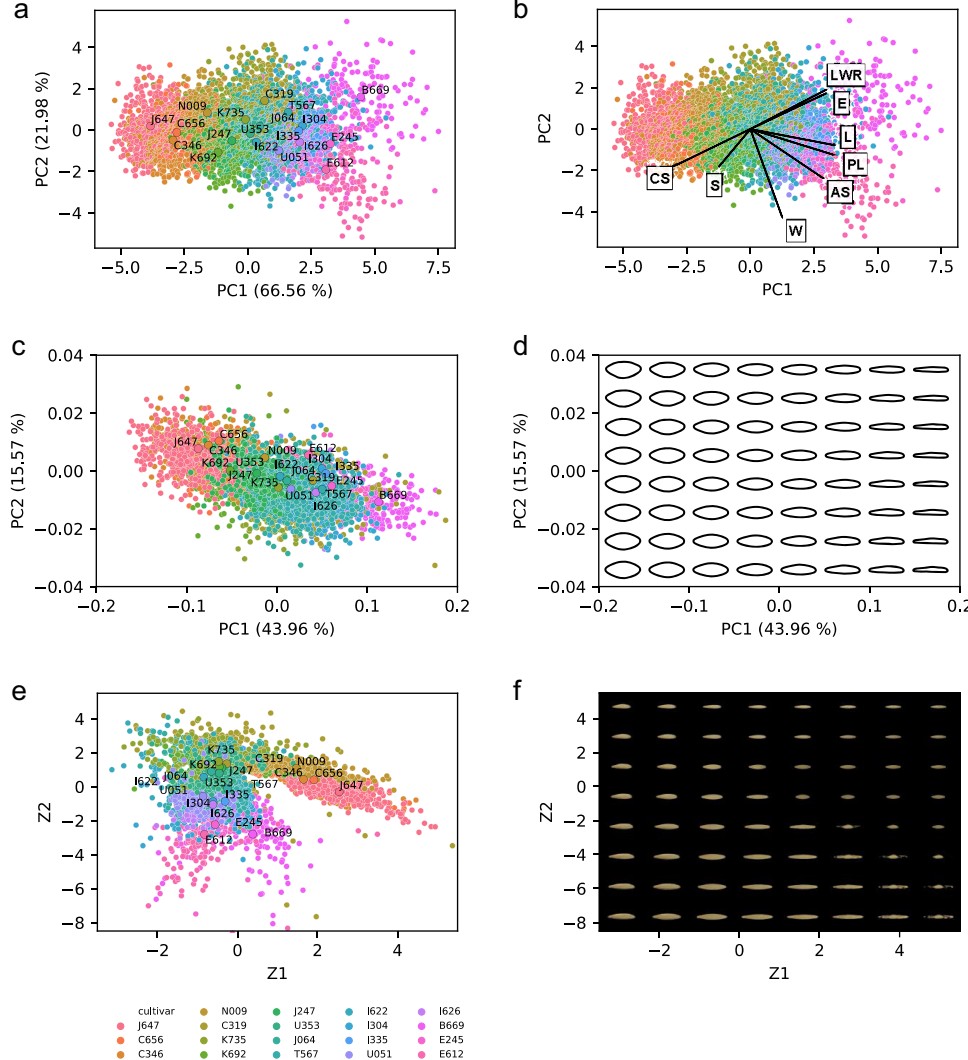

**Fig. 5 Multivariate analysis of barley seed morphology. a, b** Principal component analysis (PCA) with morphological parameters of barley seeds. Each point represents the data point of the respective seed. The colors correspond to those defined in the color legend displayed below (**e**). Mean PC1 and PC2 values of each cultivar are plotted as a large circle with text annotations in (**a**). Eigenvectors of each descriptor are drawn as arrows in (**b**). LWR length-to-width ratio, E eccentricity, L seed length, PL seed perimeter length, AS seed area, W seed width, S solidity, CS seed circularity. **c, d** PCA with elliptic Fourier descriptors (EFD). The colors and points annotated of (**c**) follow those of (**a**). Interpolation of the latent space followed by reconstruction of the contours are displayed in (**d**). **e, f** Latent space visualization of variational autoencoders (VAE). The colors and points annotated of (**e**) follow those of (**a**). Interpolation of the latent space followed by image generation using the generator of VAE are displayed in (**f**).

## Methods

**Plant materials**. Barley seeds used in this research are 19 domesticated barley (*Hordeum vulgare*) accessions and one wild barley (*H. spontaneum*) accession: B669, Suez (84); C319, Chichou; C346, Shanghai 1; C656, Tibet White 4; E245, Addis Ababa 40 (12-24-84); E612, Ethiopia 36 (CI 2225); I304, Rewari; I335, Ghazvin 1 (184); I622, H.E.S. 4 (Type 12); I626, Katana 1 (182); J064, Hayakiso 2; J247, Haruna Nijo; J647, Akashinriki; K692, Eumseong Covered 3; K735, Natsu-daikon Mugi; N009, Tilman Camp 1 (1398); T567, Goenen (997); U051, Archer; U353, Opal; H602, wild barley. All the details of the said cultivars can be obtained at the National BioResource Project (NBRP) (https://nbrp.jp). Meanwhile, seeds of rice (*Oryza sativa*, cv. Nipponbare), oat (*Avena sativa*, cv. Negusaredaiji), lettuce (*Lactuca sativa*, cv. Great Lakes), and wheat (*Triticum aestivum* cv. CS, Chinese Spring; N61, Norin 61; AL, Arina (ArinaLrFor))[51]; and Syn01, a synthetic hexaploid wheat line Ldn/KU-2076 that is generated by a cross between tetraploid wheat *Triticum turgidum* cv. Langdon and *Aegilops tauschii* strain KU-2076[52] were used in this report.

**Image acquisition**. All the barley seeds were threshed using a commercial table-top threshing system (BGA-RH1, OHYA TANZO SEISAKUSHO & Co., Japan). The seed images were captured on an EPSON GT-X900 A4 scanner with the supplied software without image enhancement. Seeds were spread uniformly on the glass, scanned at 7019 × 5100 px at 600 dpi using a blue-colored paper background.

For the image acquisition of seeds of rice, oat, lettuce, and wheat, an overhead scanner ScanSnap SV600 (Fujitsu, Japan) was used with the image size of 3508 × 2479 at 300 or 600 dpi.

**Synthetic image generation**. Single-seed images per cultivar (total of 400; 20 seed images for 20 cultivars) were isolated and saved as an individual image file. These 400 seeds were manually annotated and were used to create a non-domain -randomized training dataset used in Supplementary Fig. 3. The following describes the procedure of synthetic image generation.

First, the background regions of seed images were removed such that the pixel value other than the area of the seed will be (0,0,0) in RGB color value. As a result, 400 background-clean images were prepared to constitute a "seed image pool". For the background image, four images at the fixed size of 1024 × 1024 were cropped from the actual background used in the seed scanning process and were prepared as a "background image pool".

The synthetic image generation process is described as follows. First, an image was randomly selected from the background image pool and pasted to the virtual canvas of size 1024 × 1024. Second, another image was randomly selected from the seed image pool. Image rotation angle was randomly set upon selection. After rotation, the *x* and *y* coordinates at which the image was to be pasted were randomly determined; however, the coordinate value was restricted to a certain range so that the image does not exceed the canvas size, with which its values were

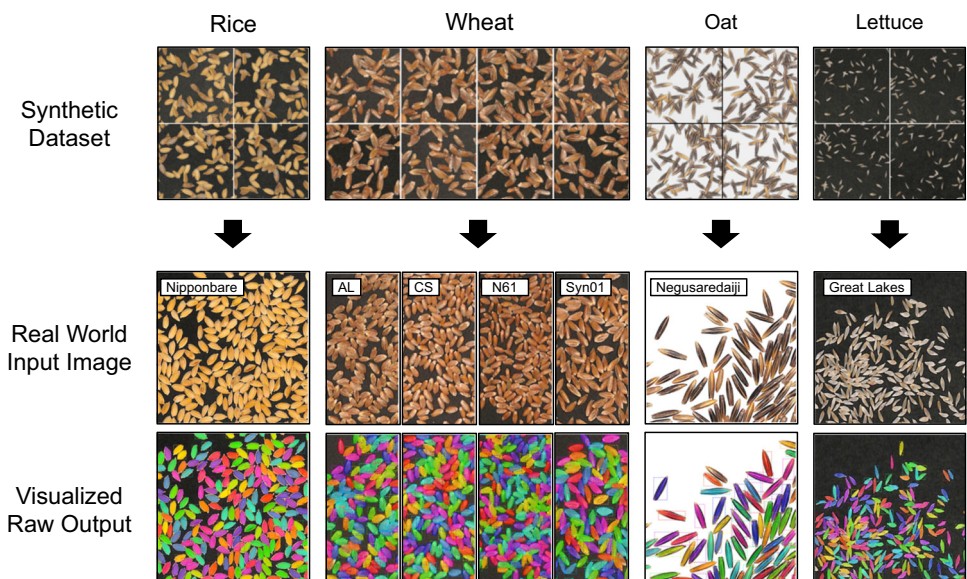

**Fig. 6 Application of our proposed pipeline to seeds of various species.** Synthetic data of the respective species were generated (top row) and the neural networks were independently trained. The inference results against the real-world input images (middle row) were visualized (bottom row). The name of the cultivar per species is overlaid, respectively.

dependent on the selected seed image size and its rotation angle. Third, the seed image was pasted to the canvas according to the determined values described above. When pasting, alpha masks were generated and utilized in alpha blending such that the area outside of the seed will be transparent and does not affect the canvas image. Moreover, utilizing the alpha mask, the seed perimeter was Gaussian blurred to decrease the artifacts resulting from the background removal process of the seed image. Notably, if the region where the image was to be pasted in the canvas already had a seed image, the overlapping proportion of the area of the seeds was calculated. If the calculated value exceeded the ratio of 0.25, pasting was canceled, and another coordinate was chosen again. The threshold percentage of overlap was arbitrarily determined based on the actual seed overlap that was observed in actual situations. A maximum of 70 pasting trials were performed to generate a single image.

During the synthetic image generation, a mask that has the same image size as the synthetic image was created by first creating a black canvas and coloring the seed region with unique colors based on the coordinate of the placing object. The coloring was performed when the seeds were randomly placed in the synthetic image. If a seed to be placed were overlapping an existing seed, the colors in the corresponding region in the mask image were replaced by the foreground color.

The above procedure generates an image size of $1024 \times 1024$ with seeds randomly oriented inside the canvas image. While in real-world images, seeds that are adjacent to the border of the image are cut off. To replicate such a situation, the borders of synthetic images were cropped to obtain the final image. The generated synthetic dataset constitutes 1200 set of data pairs of synthetic and mask images, in which each image has a size of $768 \times 768$ that was used for neural network training.

**Model training**. We used a Mask R-CNN[7] implementation on the Keras/Tensorflow backend (https://github.com/matterport/Mask_RCNN). Configuration predefined by the repository was used, including the network architectures and losses. The residual network ResNet101[53] was used for the feature extraction. From the initial weights of ResNet101 obtained by training using the MS COCO dataset, we performed fine-tuning using our synthetic seed image dataset for 40 epochs by stochastic gradient descent optimization with a learning rate of 0.001 and batch size of 2. Within the 1200 images of the synthetic dataset, 989 were used for training, 11 for validation, and 200 for the synthetic test dataset. No image augmentation was performed during training. The synthetic training data have a fixed image size of $768 \times 768$; however, the input image size for the network was not exclusively defined such that variable sizes of the image can be fed upon inference. The network outputs a set of bounding boxes and seed candidate mask regions with a probability value. A threshold value of 0.5 was defined to isolate the final mask regions.

**Real-world test dataset for model evaluation**. While the synthetic test dataset was generated according to the method described in the previous section, we prepared a real-world test dataset consisting of 20 images with which each image contained seeds derived from a homogeneous population (Fig. 2a). Each image had a size of $2000 \times 2000$. $AP_{50}$, $AP_{75}$, and AP@[.5:.95] per image (cultivar), as well as the mean AP of all images, was calculated. As the seeds to be detected per image

average to ~100 objects per image and images themselves were acquired under the same experimental condition, we used one image per cultivar for model evaluation. Ground-truth label of real-world test dataset was manually annotated with Labelbox[54]. For reference, we also prepared 200 synthetic images for testing (synthetic test dataset), which were not used for the model training or validation.

**Metrics for model evaluation**. To assess the accuracy of object detection using Mask R-CNN, we evaluated using two metrics, which were also used in the evaluation of the original report[7]. While they are commonly used measures in object recognition and instance segmentation, such as in MS COCO[15] and Pascal VOC[55] dataset, we briefly recap our evaluation metrics for clarity. During the experiment, the evaluation metrics were calculated using the Mask R-CNN distribution.

Recall: We first measured the recall, which evaluates how well the objects (i.e., seeds) are detected, which can be obtained by the ratio of true positive matches over the total number of ground-truth objects. To calculate the recall values, we determined the correct detection when the detection threshold of the intersection-over-union (IoU) between the ground-truth and predicted bounding boxes is over 0.5 (Fig. 7a). In other words, for each ground-truth bounding box, if a detected bounding box overlaps over 50%, it was counted as the true positive. Hereafter, we denote the recall measures as $Recall_{50}$.

Average precision (AP) using mask IoUs: The drawbacks of the recall measure include penalizing the false-positive detections and evaluating using the overlaps of bounding boxes that are poor approximation of the object shape. We, therefore, calculated the average precision (AP) using mask IoUs, which can be a measure of the detection accuracy (in terms of both recall and precision) as well as providing a rough measure of mask generation accuracy. During the computation of APs, we first compute the IoU between the instance masks (mask IoU), as shown in Fig. 7a. AP can be obtained based on the number of correct (i.e., true positive) and wrong (i.e., false positive) detection determined using a certain threshold of mask IoUs. Figure 7b summarizes the computation of the AP. We sort the detected instances using the class score (i.e., the confidence that the detected object is a seed, in our case) in the descending order. For the nth instance, the precision and recall, based on the mask IoU threshold, are calculated for the subset of instances from 1st to nth detections. By repeating the process for each of the instances, we obtain a receiver-operating characteristic (ROC) curve shown in Fig. 7b. The AP is defined as the ratio of the rectangle approximations of the area under the curve (AUC), which is shown as the area marked by slanted lines in the figure. APs thus takes the value from 0.0 to 1.0 (i.e., 100%). We evaluated APs using multiple mask IoU thresholds. $AP_{50}$ and $AP_{75}$ are computed using the mask IoU threshold of 0.5 and 0.75, respectively. $AP_{75}$ becomes a stricter measure than $AP_{50}$, because $AP_{75}$ requires the correct matches with more accurate instance masks. Similar to MS COCO evaluation, we also measured AP@ [.5:.95], which is the average value of APs with IoU thresholds from 0.5 to 0.95 with the interval of 0.05.

**Quantification of seed morphology**. The main application of the seed instance segmentation is to quantify phenotypes of seeds for analyzing and comparing morphological traits. In the mask image, morphological variables of seed shape such as area, width, and height were calculated using the measure.regionprops

**Fig. 7 Evaluation metrics for object detection accuracy. a** The intersection-over-union (IoU) definitions for bounding boxes and masks. **b** The average precision (AP) defined as the area under the curves (AUC), shown as the area marked with slanted lines.

module of the scikit-image library, respectively. To analyze the characteristics of seeds across different cultivars, principal component analysis (PCA) was applied to the variables. In the "Results" section, we briefly present the analysis using different types of descriptors, computed by elliptic Fourier descriptors (EFD) and variational autoencoder (VAE), both of which are described below.

Post processing: selection of isolated seeds: The instance segmentation network outputs a set of bounding boxes and seed area candidates as mask images, where some seeds overlap with each other. To analyze the seed morphology (or use for further phenotyping applications), it is required to select the seeds that are isolated (i.e., not partly hidden) from the neighboring seed instances. To select such seeds, the post-processing step was introduced. First, the bounding box coordinates were checked whether it resides inside the 5 px margin of the image. The bounding boxes that protrude the margin were removed. Second, using the solidity (ratio of the region of interest area against its convex hull area) of the respective mask as a metric, the 25% lower quantile threshold was determined and used to remove the outliers. Similarly, further outliers were removed by a 5% lower and 95% higher quantile threshold of length-to-width ratio. The threshold was empirically determined during the analysis.

Elliptic Fourier descriptors (EFD): EFD[56] has been used to quantify the contour shape of seeds[38], which approximate the contour shape as the set of different ellipses. During the computation of EFD, segmented seed images were first converted to binary mask image where the background pixel value was 0 and the seed area is 1. Next, the contour of the seed was detected by the find_contours module of the scikit-image library. The detected contours were converted to EFD coefficients using the elliptic_fourier_descriptors module of pyefd library (https://github.com/hbldh/pyefd) under the condition of harmonics 20 and with normalization so as to be rotation- and size-invariant. The output was flattened, which converted the shape of the array from $4 \times 20$ to 80. As the first three coefficients are always or nearly equal to 1, 0, 0 due to the normalization process, they were discarded upon further analysis. A total of 77 variables were used as descriptors for principal component analysis (PCA).

Variational autoencoder (VAE): Autoencoder (AE) is a type of neural network with an encoder–decoder architecture that embeds a high-dimensional input data (e.g., images) to a low-dimensional latent vector, to correctly decode the input data from the low-dimensional vector. Variational autoencoder (VAE)[48] is a variant of AE, where the distribution in the latent space is generated to fit a prior distribution (e.g., Gaussian distribution, N(0,1)). In a generative model, the low-dimensional parameters in the latent space are often used as the nonlinear approximation (i.e., dimensional reduction) of the dataset. Similar to other approximation methods like PCA, the parameters in the latent space estimated by VAE can be used for interpolation for the data distribution; the input data with different characteristics (e.g., different species) are often well separated in the space[57] compared with the conventional methods (e.g., PCA), without using the ground-truth labels for the classes during the training. We used a VAE with a CNN-based encoder–decoder network to visualize the latent space. In brief, the network receives an RGB image that has a shape of $256 \times 256 \times 3$. For the encoder, input data were first passed through four layers of convolution with filter numbers of 32, 64, 128, and 256, respectively. Since we fit the latent space to the Gaussian distribution, the log variance and the mean of the latent space are computed after full-connection layers. For the decoder, the output of the encoder was passed through four layers of deconvolution with filter numbers of 256, 128, 64, and 32, respectively. Finally, the convolution layer with three filters was added to convert the data back to an RGB image with its shape identical to the input image. In our analysis, we utilized the two-dimensional latent space (i.e., the final output of the encoder of VAE) to visualize the compressed features of the input image.

**Statistics and reproducibility**. Numbers of barley seeds analyzed per cultivar for evaluation of seed morphology in this study are as follows: 157, B669; 353, C319; 395, C346; 208, C656; 143, E245; 159, E612; 207, I304; 223, I335; 245, I622; 169, I626; 300, J064; 189, J247; 351, J647; 267, K692; 279, K735; 264, N009; 219, T567; 196, U051; 140, U353. R (ver. 3.5.1) was used for ANOVA and Tukey post hoc HSD test analysis to evaluate the statistical differences of their morphological parameters.

**Software libraries and hardware**. Computational analysis in this study was performed using Python 3.6. Keras (ver. 2.2.4) was also used with Tensorflow (ver. 1.14.0) backend for deep-learning-related processes. Single GPU (Geforce GTX 1080 Ti, NVIDIA) was used for the model training. Each epoch in training took about 186 s. For inference, an average of 3.9 s was required per image to process the real-world test dataset. OpenCV3 (ver. 3.4.2) and scikit-image (ver. 0.15.0) were used for operations in morphological calculations of the seed candidate regions as well as basic image processing. A single GPU was used for network training and inference.

**Reporting summary**. Further information on research design is available in the Nature Research Reporting Summary linked to this article.

## Data availability

Synthetically generated and real-world datasets can be obtained from the following GitHub repository (https://github.com/totti0223/crop_seed_instance_segmentation).

## Code availability

Code to reproduce the deployment of the trained Mask R-CNN and multivariate analysis is formatted as IPython notebooks and can also be obtained from the GitHub repository (https://github.com/totti0223/crop_seed_instance_segmentation). Other data and information regarding the paper are available upon reasonable request.

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

## Acknowledgements

We thank Labelbox for providing the access for the academic usage of dataset labeling. We thank Ms. Yoko Tomita at Nagoya University for assistance in the labor-intensive annotation to generate a ground-truth test dataset. We also thank Dr. Miya Mizutani for a comprehensive discussion and critical reading of the paper. The graphical abstract in Fig. 1 was rendered by Dr. Issey Takahashi who is a member of the Research Promotion Division in ITbM of Nagoya University. Dr. Shunsaku Nishiuchi provided Nipponbare rice seeds used in this study. Dr. Toshiaki Tameshige amplified and provided wheat seeds. Dr. Kentaro Shimizu amplified and provided wheat Arina seeds and Drs. Shigeo Takumi and Yoshihiro Matsuoka established, amplified, and provided synthetic wheat Ldn/KU-2076 (Syn01) seeds. This work was supported by Japan Science and Technology Agency (JST) PRESTO [Grants nos. JPMJPR17O5 (Y.T.) and JPMJPR17O3 (F.O.)], JST CREST [Grant Number JPMJCR16O4 (H.T., D.S., and S.O.)], MEXT KAKENHI [Numbers 16H06466 and 16H06464 (H.T.), 16KT0148 (D.S.), and 19K05975 (J.I.)], and JST ALCA [Number JPMJAL1011 (T.K.)]. All the barley materials are provided by the National BioResource Project (NBRP: Barley).

## Author contributions

Y.T. directed and designed the study, wrote the program codes, generated the synthetic test dataset, and performed the experiments with assistance from F.O., H.T., D.S., and K.T. H.T. and D.S. collected and scanned the barley seed images, and J.I. collected wheat images. Y.T. annotated the test dataset. Y.T., H.T., and D.S. were involved in the conceptualization of this research. Y.T., F.O., and H.T. wrote the paper with assistance from D.S., K.T., J.I., and S.O., furthermore with verification of scientific validity from all the coauthors.

## Competing interests

The authors declare no competing interests.
