## [Peer Review File · Communications Biology]

Reviewers' comments:

Reviewer #1 (Remarks to the Author):

The manuscript describes the use of domain randomization for generating training images with the goal of performing instance segmentation on images of seed trays. The resulting seed areas are used to extract morphological measurements which are then analyzed using PCA. The authors report segmentation and detection results similar to those using manual annotation.

The manuscript is well-written, with only a few minor grammatical errors.

Minor points:

- The authors' definition of deep learning is under-specified (Line 33). For example, training a simple two-layer MLP is not considered "deep learning".
- The authors' claim that AlexNet surpassed human performance in 2012 is not true (Line 36). The top-1 error of AlexNet was ~18% compared to ~5% for humans.
- Missing a word ("is"?) (Line 77), and another one ("the"?) (Line 79).
- The introduction seems to introduce two separate problems, but then refers to them interchangeably. The domain adaptation problem (the one addressed in Giuffrida et al.) is concerned with training a model on one dataset and using it to perform inference on a different dataset drawn from a completely different distribution. Note that this is not necessarily done via transfer learning/fine tuning as stated on Line 61 (for example, consider Giuffrida et al.). The other problem (the one addressed in the present manuscript) is data augmentation, where a single dataset is sampled artificially, for example, via domain randomization.
- The authors missed a very relevant previous work regarding domain randomization in plants [1].
- I believe that one important factor not described is the quality and quantity of the bank of seed images used for generating the training images. Presumably the range of variation in these images needs to capture as much of the variance in the dataset as possible? What happens when you randomly sample subsets from your seed bank? Do your results change significantly? What if you vary the size of the seed bank, is there a size which is too small?
- While it is a fun novelty, I don't think that the VAE results contribute anything to the analysis and I would consider removing them. I leave the decision to the authors.

In total, I believe that domain randomization is better on average than manual annotation and I find the segmentation results to be compelling. The PCA results for the barley cultivars are interpretable and intuitive. Although the application of domain randomization is not new in plants [1], this is the first application to seed trays that I am aware of, so I believe the application is novel enough to justify publication.

[1] Kuznichov, D., et al. (2019). Data augmentation for leaf segmentation and counting tasks in rosette plants. Proceedings of the IEEE Conference on Computer Vision and Pattern Recognition

Workshops, 2019.

Reviewer #2 (Remarks to the Author):

This was a very nice paper which demonstrated a semi-synthetic data approach for training CNNs for object detection and segmentation (plant seeds), as well as the use of autoencoders to more effectively summarize the ways seeds can differ in shape than the current proxies which must necessarily be used by biologists conducting hand measurements (kernel width/height/area). They show the same semi-synthetic approach they employ in barley can we used to train separate models for the seeds of a range of other crop species. They also have made their code and training testing data available through github which is incredibly important for this work to have real work impacts for plant biologists who likely wouldn't be able to rebuild the methods described here based solely on the summaries of the algorithms traditionally provided in methods sections.

I have one moderate concern and two minor suggestions.

Moderate concern:

1. The authors show that their semi-synthetic training data, generated from hand segmentation of 400 barley kernels/grains, performs quite well in segmenting additional real world images. Having not worked directly in seed segmentation, is there any benchmark they can compare to to estimate how well a network trained directly on 400 hand annotated barley kernels/grains would perform at the same test data? It's clear they have a good tool that does its job well, but it was harder for me to judge how much of that value comes from the semi-synthetic training data strategy and how much is just what would be expected from a 400 grain training dataset in the task of identifying grains against a largely uniform background.

Minor

2. I think the authors may be over selling the role seed size plays in determining yield:
Line 93: I would suggest to the authors that they omit this sentence. Most plant biologists will have an argument for why the trait or traits they study are the "most important."
Line 93-94: This citation deals with a single gene which effects seed mass, but not seed shape. Seed size, and to a lesser extent shape, are definitely agriculturally important phenotypes. They are one of a number of yield components and the one which appears to be influenced by the environment the latest in the growing season. Variation in seed size/shape also has predictive power on germination rates (Revilla et al doi: 10.2135/cropsci1999.0011183X003900020007x) and development (Elwell et al doi: 10.1111/j.1365-3040.2010.02243.x). It's quite possible for phenotyping kernels/seeds to be a worthy goal without arguing that is is the single most important predictor of yield. At least in maize, most of the increase in yield in the past century has come from the development of varieties that produce greater numbers of kernels per plant and tolerate greater numbers of plants per mu/hectare/acre, not larger individual kernels.

3. Lines 104-112: This is a good review of some of the previous work in automated seed phenotyping from images, but excludes maize work. I'd definitely suggest including Miller et al (doi: 10.1111/tpj.13320) and possibly Wen et al (doi: 10.15258/sst.2015.43.1.07)

Reviewer #3 (Remarks to the Author):

This paper presents an approach to instance segmentation of seeds. Overall this is a well written paper. It presents some interesting ideas and the implementation is well thought out.

The synthetic image generation route is a natural approach to this problem. Data can be collected easily in such a way that makes it easy to automatically annotate the data for more difficult problems. The method describe is fine and the use of scanner is smart, providing an orthographic view of the seeds. I believe there may be some room for improvement in terms of data generation though. I think much more could have been done to these seeds during the generation to make them appear more different. From the prose, I feel as though only rotation was performed to the seeds. There was room also for adjusting scale and colour information. This may result in a more robust network over all.

My one concern about this paper is that the first few pages give the impression that it will be suitable for overlapping instance segmentation problems. I am skepticle of this claim for a couple of reasons. First, Mask-RCNN struggles, in general, with overlapping instances. The bounding box detection will most likely absorb one two bounding boxes into one, especially if they are of the same class and one is occluded quite heavily. Second, on line 175, it is mentioned that during generation, if there is more than 25% overlap, the pasting will be cancelled and a new location will be generated. If the method were really capable of overlapping instance segmentation, this would not be a primary concern during data generation.

Replicating the borders being cut off during data generation, by later cropping the images, is a good idea, and avoids the network from being confused in future when it encounters a seed which is cut in half. I am curious to understand however why such a large spatial resolution was used for the input. Are the individual seeds very small?

I have a minor concern about the testing set, which is that the synthetic testing images were generated from the same set of automatically extracted seeds already used for the training set generation? If this is the case, your results could be unfairly biased towards higher accuracy.

I am a little confused about the Variational Autoencoder section (277). Perhaps I have missed it, but I am not sure why we ware discussing this? Some connecting prose would really help the reader follow this.

There are a few questions which I think would be nice to answer at some section in the paper, such as the time it takes to train the network? How long does a single forward pass through the network take? What model of GPU was used for training?

Finally, I really enjoyed the discussion of elliptic fourier descriptors and PCA in the supplementary material. This was a very nice addition to the paper.

Some more detailed notes:

- I feel there is room for improvement in the definition of deep learning.
- Presumably CG (line 65, 74 etc) refers to Computer Graphics? Could you include this?
- Sentence seems a little malformed on line 120.

Dear Reviewer #1, #2, and #3

We appreciate the valuable and constructive comments from all the reviewers. In particular, we are grateful that the reviewers are interested and agreed on the importance of our study. Point-by-point response according to the reviewers' comments are given below. We have reflected the answers to the revised version of the manuscript. The modified or added parts are highlighted in red in the revised manuscript except for citation numbers, as majority of the numbering has been changed upon revision.

Replies to Reviewer #1

Comment 1-1: The authors' definition of deep learning is under-specified (Line 33). For example, training a simple two-layer MLP is not considered "deep learning".

Comment 1-2: The authors' claim that AlexNet surpassed human performance in 2012 is not true (Line 36). The top-1 error of AlexNet was ~18% compared to ~5% for humans.

Reply to 1-1 and 1-2: We first thank reviewer 1 for the extensive review. As in the reviewer's comment, the description and the historical fact about the deep neural network was inaccurate on the following points:

- The definition of deep learning does not basically include the simple/multi-layer perceptrons of the number of layers < 4 .
- The part "*namely AlexNet, outperformed the human image classification accuracy to classify 1000 categories*" was inappropriate. The human baseline was 5% then, and the year that the NN outperformed that accuracy for ImageNet was in 2015 (w/ ResNet-based network).

Rather giving the definition of deep learning or introducing the history, the goal of the paragraph was intended to introduce that deep learning is applied for various image analysis tasks, such as semantic/instance segmentation. Therefore, we have largely simplified the paragraph to focus the essential things by omitting the historical cues (line 33 - 36). We believe this have greatly improved the clarity.

Comment 1-3: Missing a word ("is"?) (Line 77), and another one ("the"?) (Line 79).

Reply to 1-3: We have revised the manuscript according to your suggestion.

Comment 1-4: The introduction seems to introduce two separate problems, but then refers to them interchangeably. The domain adaptation problem (the one addressed in Giuffrida et al.) is concerned with training a model on one dataset and using it to perform inference on a different dataset drawn from a completely different distribution. Note that this is not necessarily done via transfer learning/fine tuning as stated on Line 61 (for example, consider Giuffrida et al.). The other problem (the one addressed in the present manuscript) is data augmentation, where a single dataset is sampled artificially, for example, via domain randomization.

Reply to 1-4: As pointed, we introduced domain adaptation and data augmentation interchangeably, which may have made the readability of the introduction section difficult. The main reason was we defined our work from the viewpoint of domain randomization, which falls in a category of sim2real transfer techniques.

Sim2real---using synthetic training images for real tasks---fundamentally or potentially involves a domain adaptation problem. Although some techniques directly use synthetic images (e.g. Varol *et al*), a straightforward idea to do sim2real transfer is to use domain adaptation techniques that *move* the feature distribution, such as by making synthetic images realistic (e.g. Shrivastava *et al.*) or using Cycle-GAN-based approaches to learn domain-consistent features [a]. Meanwhile, domain randomization can be categorized as another technique to solve sim2real problem by (some kind of) data augmentation, to *cover* the feature distribution of real scenes by creating synthetic images using various (randomized) parameters.

In the original manuscript, the above discussion was unclear and confusing. We have simplified the part by focusing on the techniques using synthetic images. In the revised the manuscript (page 2 to 3),

- we first introduce approaches using synthetic images as training dataset
- then we discuss the data creation with realistic synthesis (such as using GANs)
- and we finally introduce the domain randomization approaches to increase the variation of synthetic dataset.

We hope the modified description concisely introduce the works related to sim2real approaches.

[a] CyCADA: Cycle-Consistent Adversarial Domain Adaptation, Judy Hoffman et al. ICML2018.

Comment 1-5: The authors missed a very relevant previous work regarding domain randomization in plants [1]. [1] Kuznichov, D., et al. (2019). Data augmentation for leaf segmentation and counting tasks in rosette plants. Proceedings of the IEEE Conference on Computer Vision and Pattern Recognition Workshops, 2019.

Reply to 1-5: Thank you for your critical comment. We have added the citation in the introduction section as bellow.

(line 78 - 81)

“In the plant phenotyping field, recently, Kuznichov et al. proposed a method to segment and count leaves of not only *Arabidopsis*, but also that of avocado and banana, by using a synthetic leaf textures located with various size/angles, so as to mimic images that were acquired in real agricultural scenes”.

Comment 1-6: I believe that one important factor not described is the quality and quantity of the bank of seed images used for generating the training images. Presumably the range of variation in these images needs to capture as much of the variance in the dataset as possible? What happens when you randomly sample subsets from your seed bank? Do your results change significantly? What if you vary the size of the seed bank, is there a size which is too small?

Reply to 1-6: We agree with the reviewer that investigating the relationship between the parameters that affect synthetic data variance and the model performance indeed provide readers further

information. However, upon preliminary model training, we experienced that such parameters (e.g. number of seed image per cultivar, number and resolution of synthetic training images, and sampling patterns from seed banks) greatly vary based on the combination and numbers of cultivars we want to analyze. In our experiment, we empirically used the manual segmentation of 20 seeds per cultivar with the same resolution as the real-world image and achieved good segmentation results, based on the augmented version of the dataset. Since creating the manual segmentation of 20 seeds does not require a notable time, in the practical viewpoint, we have thought that it may be not quite important to evaluate such parameters for manual segmentation. Nonetheless, given that such discussion is important to state, we added further discussion to address your comment, regarding the importance of quality and quantity of the seed images as below.

(line 486 - 494)

“Notably, it is expected that the model performance will be greatly influenced by the image resolution and variance of seed images used to create the synthetic image, as well as the number of images that constitute the training dataset. Optimal parameters of such will also depend on the type of cultivars that constitute the test dataset. In this study, we used a fixed condition for synthetic dataset generation, in order to prioritize to demonstrate the effectiveness of domain randomization for seed phenotyping. However, in practical situations where respective users build and execute a customized pipeline, parameter search may benefit them by providing minimal dataset requirement that leads to calculation cost efficiency. Moreover, introducing additional image augmentation techniques in the synthetic dataset such as random color shift and zoom will lead to a more robust model.”

Comment 1-7: While it is a fun novelty, I don't think that the VAE results contribute anything to the analysis and I would consider removing them. I leave the decision to the authors.

Reply to 1-7: Thank you for your comment. We understand your opinion, however as the other authors pointed out their interest, we will leave in the results for further reference for the field of art.

Replies to Reviewer #2

Comment 2-1: The authors show that their semi-synthetic training data, generated from hand segmentation of 400 barley kernels/grains, performs quite well in segmenting additional real world images. Having not worked directly in seed segmentation, is there any benchmark they can compare to to estimate how well a network trained directly on 400 hand annotated barley kernels/grains would perform at the same test data? It's clear they have a good tool that does its job well, but it was harder for me to judge how much of that value comes from the semi-synthetic training data strategy and how much is just what would be expected from a 400 grain training dataset in the task of identifying grains against a largely uniform background.

Reply to 2-1: Thank you for your suggestion. According to your comment, we have trained the network directly with 400 hand annotated barley kernels without any domain randomization. Please see the newly added Figure S3 for visual details. Briefly, the network hardly learned to segregate and

locate seeds durable for phenotyping, despite of the identical training parameter as the initial training condition. This was especially apparent with the seeds that were physically touching each other. Since such obvious difference, we have not decided to evaluate its model performance. We have added such result in the manuscript as bellow. I hope this should further highlight the power of domain randomization we have adopted.

(line 349 - 352)

“It is noteworthy that when the Mask R-CNN model was trained with manually annotated seeds, the network showed poor performance in segmenting the seed regions (Fig. S3). This was especially apparent when the seeds were physically touching each other and forming a dense cluster, which further support the efficiency of domain randomization”.

Comment 2-2: Line 93: I would suggest to the authors that they omit this sentence. Most plant biologists will have an argument for why the trait or traits they study are the “most important.”

Reply to 2-2: We agree with the reviewer, and have omitted this sentence in the text.

Comment 2-3: Line 93-94: This citation deals with a single gene which effects seed mass, but not seed shape. Seed size, and to a lesser extent shape, are definitely agriculturally important phenotypes. They are one of a number of yield components and the one which appears to be influenced by the environment the latest in the growing season. Variation in seed size/shape also has predictive power on germination rates (Revilla et al doi: 10.2135/cropsci1999.0011183X003900020007x) and development (Elwell et al doi: 10.1111/j.1365-3040.2010.02243.x). It’s quite possible for phenotyping kernels/seeds to be a worthy goal without arguing that is is the single most important predictor of yield. At least in maize, most of the increase in yield in the past century has come from the development of varieties that produce greater numbers of kernels per plant and tolerate greater numbers of plants per mu/hectare/acre, not larger individual kernels.

Reply to 2-3: We thank the reviewer for these noteworthy and insightful remarks on seed size and shape. As the reviewer suggests, seed size and shape are important phenotypes. We have added following sentences to the text.

(lines 83 - 87)

“Seed shape, along with seed size, is an important agricultural phenotype. They consist of yield components of crops which is affected by environmental condition in the later developmental stage. The seed size and shape can be predictive on germination rates and subsequent development of plants (refs: Revilla et al 1999, Elwell et al. 2010). Genetic alteration of seed size contributed a significant increase in thousand-grain weight in contemporary barley cultivated germplasm (Sakuma et al. 2017)”

Comment 2-4: Lines 104-112: This is a good review of some of the previous work in automated seed phenotyping from images, but excludes maize work. I’d definitely suggest including Miller et al (doi: 10.1111/tpj.13320) and possibly Wen et al (doi: 10.15258/sst.2015.43.1.07)

Reply to 2-4: Thank you for the comment. We have summarized the suggested citations with the following addition to the text.

(lines 103 - 106)

“Miller et al reported a high-throughput image analysis that measure morphological traits of maize ears, cobs and kernels. Wen et al. developed an image analysis software that can measure seed shape parameters such as width, length and projected area, as well as the color features of maize seeds: they found a correlation between these physical characteristics with seed vigour.”

Replies to Reviewer #3

Comment 3-1: I feel as though only rotation was performed to the seeds. There was room also for adjusting scale and colour information. This may result in a more robust network over all.

Reply to 3-1: As the reviewer pointed out, we adopted only the rotation and random pasting for generating the synthetic dataset, however it made the model performance sufficient for practical usage in the current dataset (0.95 AP50 for the Real World test dataset). Nonetheless, we agree adding further augmenting technique such as scaling and colour transformation will make the model more robust to various situations, including cultivars not included in the training dataset (in other words, extrapolation). We added such discussions in the manuscript along with image quality and dataset variation issues pointed out by reviewer 1. Please see **Reply to 1-6** for details.

Comment 3-2: My one concern about this paper is that the first few pages give the impression that it will be suitable for overlapping instance segmentation problems. I am scepticle of this claim for a couple of reasons. First, Mask-RCNN struggles, in general, with overlapping instances. The bounding box detection will most likely absorb one two bounding boxes into one, especially if they are of the same class and one is occluded quite heavily. Second, on line 175, it is mentioned that during generation, if there is more than 25% overlap, the pasting will be cancelled and a new location will be generated. If the method were really capable of overlapping instance segmentation, this would not be a primary concern during data generation.

Reply to 3-2: As pointed, Mask R-CNN often misses hardly occluded instances due to the non-maximum suppression (NMS) process, which remove the bounding boxes overlapping over a designated IoU threshold, after region proposal generation. We set the detection NMS threshold as IoU of 0.3, it would capable with mild overlap.

Our synthetic data generation only represent the scenes with “mildly” overlapping seeds, maximum of 0.25 overlap. (Although the threshold of 0.25 for the overlap of seed masks cannot be simply compared with the NMS threshold of 0.3, because it is computed for bounding boxes).

The robustness for overlapping instances is anyway largely rely on the instance segmentation method, so we may have somehow oversold the proposed method in the context of “densely overlapping objects”. We have decided not to argue the proposed method can capable to separate the overlapping seeds, although showing the results with the notation of “mildly overlapping seeds”.

We would like to note here that, as we only used isolated seed instances for the seed morphology analysis (due to the difficulties of acquiring morphological features from partly hidden instances), addressing occlusions is not the primal goal for our system. Nonetheless, segmenting severely overlapped seeds may contribute to counting objects, but are not suitable for phenotypic analysis of seeds anyway.

Comment 3-3: Replicating the borders being cut off during data generation, by later cropping the images, is a good idea, and avoids the network from being confused in future when it encounters a seed which is cut in half. I am curious to understand however why such a large spatial resolution was used for the input. Are the individual seeds very small?

Reply to 3-3: The respective size of the seeds of cultivars handled in our study were “not very small” in terms of Mask-RCNN to detect even in lower resolution (e.g. lower than half of original resolution). We used the original image resolution obtained by a commercially available scanner acquired in order to keep the pipeline simple. Moreover, shrinking the image and back to its resolution upon training and inference will require an additional step to evaluate whether that will affect the accuracy of phenotypic parameters handled later in our study. Therefore, we used such resolution since the commercially available GPU memory was capable to process even in high resolution.

Comment 3-4: I have a minor concern about the testing set, which is that the synthetic testing images were generated from the same set of automatically extracted seeds already used for the training set generation? If this is the case, your results could be unfairly biased towards higher accuracy.

Reply to 3-4: In this study, we prepared two types of test dataset. First is a synthetic dataset created by the method used to prepare the training dataset. Second is a dataset of real-world images acquired by scanners (Table 1, Synthetic Test Dataset and Real World Test Dataset, respectively). As the reviewer pointed out, we cannot exclude the possibility that the prior contain a data leak leading to biased accuracy even the orientation of seed orientation is completely independent. However, since the second real-world dataset is completely independent from the training dataset and show sufficient metric (0.95 AP50) and we value the latter for model performance and for practical usage, we currently think is sufficient for now. Nonetheless, we added supplemental explanation for further clarity (line 344 - 347).

Comment 3-5: I am a little confused about the Variational Autoencoder section (277). Perhaps I have missed it, but I am not sure why we were discussing this? Some connecting prose would really help the reader follow this.

Reply to 3-5: Characterization of seed phenotype greatly varies on the predefined parameters we handle prior to analysis. We’ve envisioned that leaving such phenotypic parameters (feature representations) to the neural networks may overcome such limitations. Such trend is demonstrated by the report by Ubbens et al., 2019*, which they input all of the plant image data for clustering, enabling to obtain complex phenotypes into a latent space. Indeed our VAE also succeeded to learn and cluster

seed length and size, addition to colors which the latter we were initially unaware of. We have added a brief motive and future perspective in the results section for clarity (line 445 - 453).

.
*Jordan Ubbens, Mikolaj Cieslak, Przemyslaw Prusinkiewicz, Isobel Parkin, Jana Ebersbach, and Ian Stavness, "Latent Space Phenotyping: Automatic Image-Based Phenotyping for Treatment Studies," Plant Phenomics, vol. 2020, Article ID 5801869, 13 pages, 2020.
<https://doi.org/10.34133/2020/5801869>.

Comment 3-6: There are a few questions which I think would be nice to answer at some section in the paper, such as the time it takes to train the network? How long does a single forward pass through the network take? What model of GPU was used for training?

Reply to 3-6: Thank you for your comment. We have added such information in the Software Libraries and Hardware section of Materials and Methods.

"Single GPU (Geforce GTX 1080 Ti, NVIDIA) was used for the model training. Each epoch in training took about 186 seconds, while 3.9 seconds per image for inference."

Comment 3-7:

- I feel there is room for improvement in the definition of deep learning.
- Presumably CG (line 65, 74 etc) refers to Computer Graphics? Could you include this?
- Sentence seems a little malformed on line 120.

Reply to 3-7: We have revised the respective sections as the reviewer's comment.

REVIEWERS' COMMENTS:

Reviewer #1 (Remarks to the Author):

I believe that the authors have adequately addressed the minor concerns I had with the initial draft of the manuscript and I recommend its publication.

Reviewer #2 (Remarks to the Author):

The authors have addressed all of my comments. Thank you for your thoughtful efforts in each response.

Reviewer #3 (Remarks to the Author):

Dear authors, many thanks for making the tweaks to this paper. I am satisfied with the changes. It's a nice paper!